# Automated Gold Grain Counting. Part 1: Why Counts Matter!

Réjean Girard *, Jonathan Tremblay, Alexandre Néron and Hugues Longuépée 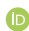

IOS Services Géoscientifiques Inc., Chicoutimi, QC G7J 3Y2, Canada; jtremblay@iosgeo.com (J.T.); neron.alex@iosgeo.com (A.N.); hlonguepee@iosgeo.com (H.L.)
* Correspondence: rejeang@iosgeo.com

**Abstract:** The quantitative and qualitative assessment of gold grains from samples of glacial till is a well-established method for exploring gold deposits hidden under glaciated cover. This method, which is widely used in the industry and has resulted in numerous successes in locating gold deposits in glaciated terrain, is still based on artisanal gravity separation techniques and visual identification. However, being artisanal, it is limited by inconsistent recoveries and difficulties associated with visually identifying the predominantly small gold grains. These limitations hinder its capacity to decipher subtle or complex signals. To improve detection limits through the recovery of small gold grains, a new approach has recently been introduced into the industry, which is commercially referred to as the "ARTGold" procedure. This procedure involves the use of an optimized miniature sluice box coupled with an automated scanning electron microscopy routine. The capabilities of this improved method were highlighted in this study by comparing till surveys conducted around the Borden gold deposit (Ontario, Canada) using the conventional and improved methods at both local and regional scales. Relative to that with the conventional approach, the improved method increased the recovery of gold grains from samples (regional and down-ice mineralization) by almost one order of magnitude. (regional and down-ice mineralization), dominantly in regard of the small size fractions. Increasing the counts in low-abundance regional samples allows for a better discrimination between background signals and significant dispersions. The described method offers an alternative for improving the characterization of gold dispersal in glaciated terrain and related gold deposit footprints.

**Keywords:** gold; till; gold grain recovery; gold grain size; automated SEM; gold exploration; drift prospecting

## 1. Introduction

Since the dawn of civilization, detrital gold has been recovered from sediments. Gold panning is a skillful art, and for this reason, artisans deserve respect. However, this is still an artisanal technique, rather than a science. Tracking the dispersion of gold grain in an alluvial system has been used for about 200 years by gold panners in search of the "mother lode". It grew as a systematic exploration method about 50 years ago, when gold grain counting started being conducted under laboratory conditions for mineral exploration companies. Since then, it has been extensively used in glaciated terrain in the exploration industry, and "Drift prospecting" has become the method of choice for grassroots gold exploration in North America and Northern Eurasia. However, the method introduced in the 1970s [1–4], which is heavily dependent on operators' skills and poorly parameterized, has barely evolved. The precepts of the method are that gold can efficiently be concentrated using the gravimetric method and that gold grains can be easily identified. We challenge both precepts in this study.

The flow of continental ice eroding bedrock produces glacial sediments that inherit the signature of local bedrock sources, including ore bodies [5]. The identification of geochemical and mineralogical dispersal trains within these glacial sediments can therefore be used in tracing mineral deposits hidden under glaciated cover. These approaches are

particularly relevant for Canada and Northern Eurasia, where most undiscovered ore deposits are likely buried under the glacial sediments that cover much of the landscape. A large number of important discoveries, such as the Douay deposit in Northeastern Québec or Rainy River mine in Western Ontario [6,7], have been made. The glacial sediment geochemistry and indicator mineral methods have proven to be efficient in drift prospecting for a wide variety of commodities [8–13], particularly diamonds [14–16]. Nonetheless, these methods are predominantly used for gold exploration, being the most efficient grassroots methods for this type of deposit. Since the bulk of investment in mineral exploration is devoted to gold (e.g., in 2017, 65% of exploration spending was devoted to gold in Canada [17]), the method is consequently of widespread usage.

While gold assaying of glacial sediments represents a far faster and cheaper approach than gold grain counting methods, the information it generates is restricted by the analytical detection limit and is typically erratic due to the nugget effect. In contrast, indicator mineral methods have virtually no detection limit, since an unlimited amount of material can be processed, and the nugget effect can be bypassed by counting discreet gold grains, regardless of their weight. Furthermore, unlike most other commodities, such as diamonds, gold itself remains the main indicator mineral for gold deposits [1,13]. In addition to grain counts, the methods can generate a combination of useful information, such as the size, shape, and composition of the grains, which may yield indications as to the type of ore deposit and transport distances [18]. In this sense, the recovery of gold grains from glacial sediments, along with their identification, counting, and characterization, offers a more integrative approach for the exploration of gold deposits in glaciated terrains.

Numerous laboratories offer gold grain counting on a commercial basis. The method used to process glacial sediments in almost all laboratories (hereafter referred as the "conventional method") is based on a combination of gravity separation techniques (mainly shaking tables and hand panning) and visual identification (typically under binocular stereomicroscopes). While this method is widely used and has led to numerous successes [2,6,7,19–27], some studies also suggest that the conventional method yields erratic results due to its low counts, since it is not effective in recovering/identifying very small gold grains [3,4,28–32]. In fact, most laboratories report only an uncertified visual grain count, with grain size estimation and simple shape interpretation [33]. Until recently, such results were taken for granted, since it was not possible to benchmark these against more sensitive methods.

Various SEM-based gold scanning techniques are available, but their usage in drift exploration is limited due to the analytical cost and representativeness issues. These methods were developed mainly for metallurgical assessment purposes, where gold grains are abundant, samples less numerous, and the budget less constrained. Routines such as RPS (rare phase search) for an MLA (Mineral Liberation Analyser, FEI, now Thermo-Fisher Scientific) [34–37], TMS (trace mineral search) for a QEMScan (Quantitative estimation of mineral by scanning, FEI, now Thermo-Fisher Scientific) [34,35,38,39], or their counterpart in other systems, such as Mineralogic (Zeiss), TIMA (Tescan Itegrated Mineral Analyser, Tescan), or AMICS (Advanced Mineral Identification and Characterization System, Bruker), can be adapted to scan heavy mineral concentrate for gold grains and are commercialized or even trademarked (e.g., "Rimscan") by some laboratories. However, an exhaustive review of the Quebec and Ontario governments' data repositories, where drift prospecting is extensively used, shows that there have been almost no assessment reports filed by the mineral industry indicating their use and that the vast majority of till surveys still rely on conventional optical sorting.

The conventional technique is currently limited by the efficiency of the procedure. Regardless of how careful the laboratory is, it is not possible to significantly improve the results beyond those achieved by the currently used method. Improving the results requires the use of more sophisticated technologies, just as the improvement of the detection limits on assays required the transition from atomic adsorption to ICP-MS. The recovery and counting of large gold grains is easy. Difficulties arise in the recovery and counting of small

grains. Therefore, the development of more efficient techniques requires that attention be paid to small grains, which means the simultaneous improvement of both the recovery and identification techniques. The development of counting techniques without the capacity to recover fine-grained gold would be meaningless and recovering small grain without being capable of counting them would be a waste of time and effort. Given that the cumulative proportion of gold grains sharply increases as the grain size decreases in both in rocks and unconsolidated sediments [40], the loss of small grains through the use of the conventional method may generate skewed data. Shelp and Nichol [3] have suggested that the conventional method fails to recover 85% to 96% of the gold grains within heavy mineral concentrates from tills of the Canadian Shield.

Improving procedures to increase and systemize the recovery of minute gold from glacial sediments and making the identification of gold grains more robust will improve the quality and dependability of results. This will, in turn, enable the optimization of exploration programs, either through an overall discovery cost reduction, an increase of the rate of discovery in difficult settings, or simply a minimization of the risk associated with decision processes. In the current contribution, we assess a method that enhances the recovery of minute gold grains and provide automated certified identification. A comparative study of results of the improved and conventional methods is presented, using the Borden Gold project (Newmont Canada, formerly Goldcorp Canada) as an example. The implications for exploration supported by statistical analysis are discussed. The use of detailed data collected from gold grains, such as their size, morphology, and chemistry, is discussed in companion papers [41,42].

## 2. Gold Grain Size Distribution in Source Rocks

Metallic gold is nearly the only form of gold in ore deposits. Despite the tremendous amount of data on gold deposits in the literature, the gold grain size distribution in mineralized samples remains poorly documented and is mostly limited to localized metallogenic studies or metallurgical testing. Most metallogenic studies are based on a limited number of grains, which were obtained from highly mineralized zones. It is uncertain if these results can be extrapolated to the entire deposit and/or to its low-grade mineralized aureole. The grain size distribution of gold from its deposit core versus its aureole has apparently not been addressed. Conversely, metallurgic studies are biased by the beneficiation process or measurement methodology, which may skew the grain size distribution. No comprehensive review of the grain size distribution in mineralized occurrences is available, and no generalization can therefore be drawn about the potential variability of gold grain distribution once they are liberated, transported, and deposited in glacial sediments.

The paucity of gold grain size distribution studies is partly due to the difficulty of acquiring such information, because the grains' smallness and erratic distribution impair such measurements. Yet, knowing how gold grain sizes are distributed in gold deposits is a premise for the optimization of metallurgical recovery, as well as the implementation of exploration programs in glaciated terrains. So far, studies reporting information on the gold grain size distribution suggest that in most deposits, gold mainly occurs as very minute grains that are difficult to measure and count. Gold placers, in which gold grains are recycled, are the exception.

In orogenic gold deposits, which represent the main source of gold production in Canada and Eurasia [43,44], most gold grains appear to be <50 μm in apparent diameter. In a study of the Kalgoorlie ores (Western Australia), the documented gold grains range from <1 to 70 μm, with most grains <35 μm [45]. Another study [46] reports that in the Red October gold project (Western Australia), gold grains ranged from <10 μm to 3000 μm, with an average of 60 μm, and that in the Vivien gold project (Western Australia), gold grains ranged from <10 μm to 2000 μm, with 50% <300 μm. In the Charters Towers gold project (Queensland, Australia), the same study also reports gold grains between <10 μm and 2000 μm, with 96% of all gold grains observed in a high-grade zone being <50 μm. In

the Cononish gold project (Scotland), the documented gold grains ranged from <10 μm to 1200 μm, with most <40 μm [46], and in the Nalunaq deposit (Greenland), 80% of the documented gold grains were <10 μm [47]. Similarly, a microscopic study of 50 Canadian orogenic gold ores [48] reported that 75% of the gold was fine grained (from 0.1 to 100 μm), with the remaining 25% of the population coarse grained (from 100 to 10,000 μm), and the most common size range of gold grains was 40–50 μm.

The gold grain size appears, in general, to control the gold grade, with low-grade ores containing mainly fine gold grains and high-grade ores containing mostly coarse gold grains [49]. This grain size–grade relation is suspected to be related to distinct paragenetic stages, with the fine-grained gold responsible for the low-grade background of the ore body and the coarse-grained gold responsible for the high-grade clusters [50]. For instance, in the sheeted vein zone of the San Antonio deposit (South America), 70% of gold grains are <50 μm in the low-grade zone, whereas only 30% of gold grains are <50 μm in the high-grade zone [51]. However, Haycock's study [48] was essentially on higher-grade gold deposits that were producers or potential producers in 1937 and could be more easily processed. Today's gold ores are more likely to be exploited at a lower grade (i.e., more difficult to process), as they are dominated by fine-grained gold.

From the grain size distribution reported by Haycok [48], it was found that the submicroscopic gold (<0.1 μm) in orogenic deposits was negligible, as opposed to Carlin-type mineralization, in which gold mainly occurs as nano-particles [52,53]. However, in other styles of gold mineralization, the gold grain sizes seem to be similar to those of orogenic deposits. In the Trout Lake VMS (Flin Flon, Manitoba, Canada), 75% of gold alloy grains are <21 μm, with a median of 11 μm [54]. A study of 323 gold grains from 89 thin sections of the Pebble porphyry (Alaska, USA) shows that 96% of them are <15 μm, with an average of 3.8 μm [55]. Similarly, the investigation of 56 gold grains of a core sample of the Grasberg porphyry (Papua, Indonesia) by high-resolution X-ray computed tomography reveals that the grains range from 7 to 43 μm [56]. Finally, it was reported that 411 Au alloy grains in a sample of the Au-horizon of the Skaergaard Complex (Greenland, Denmark) range from 1.6 to 56.9 μm, with an average of 22.6 μm [57].

Over the last 30 years, the authors have accumulated an abundance of results from ore petrography studies on a wide variety of gold occurrences located in Abitibi and James-Bay, Superior Craton, Canada [58]. A total of 4316 gold grains (Figure 1) down to 1 μm in diameter, of which 3934 were smaller than 100 μm, were measured on hundreds of thin sections from a large variety of mineral occurrences. The measurements were made with an eye-piece graticule on a petrographic microscope (Zeiss AxioImager M2m, Neofluor objectives, 1000× (Carl Zeiss A.G., Oberkochem, Germany) and Leitz Laborlux-2, Fluotar objectives, 500× or 1200× OEL (Leica Microsystems GmbH, Wetzlar, Germany)). The measurement method has been consistent over time, the measurements have been conducted by the same petrographer (Mme. Lucie Tremblay, geologist), and the smallness has been limited in terms of the optical resolution (about 0.4 μm or 1 μm grain). The measured diameter is expected to be slightly underestimated, since the polished surface of the grains does not necessarily truncate the largest portion of the grains, and no stereological correction was made [59]. It is considered that these represent an in-situ gold grain population, which is representative of Archean orogenic or intrusion-related deposits that were susceptible to erosion by glacier and dispersed in glacial sediments. Despite its numerous biases and limitations, this population can be considered as the most representative reference in a comparison of the gold grain characteristics extracted from glacial sediments.

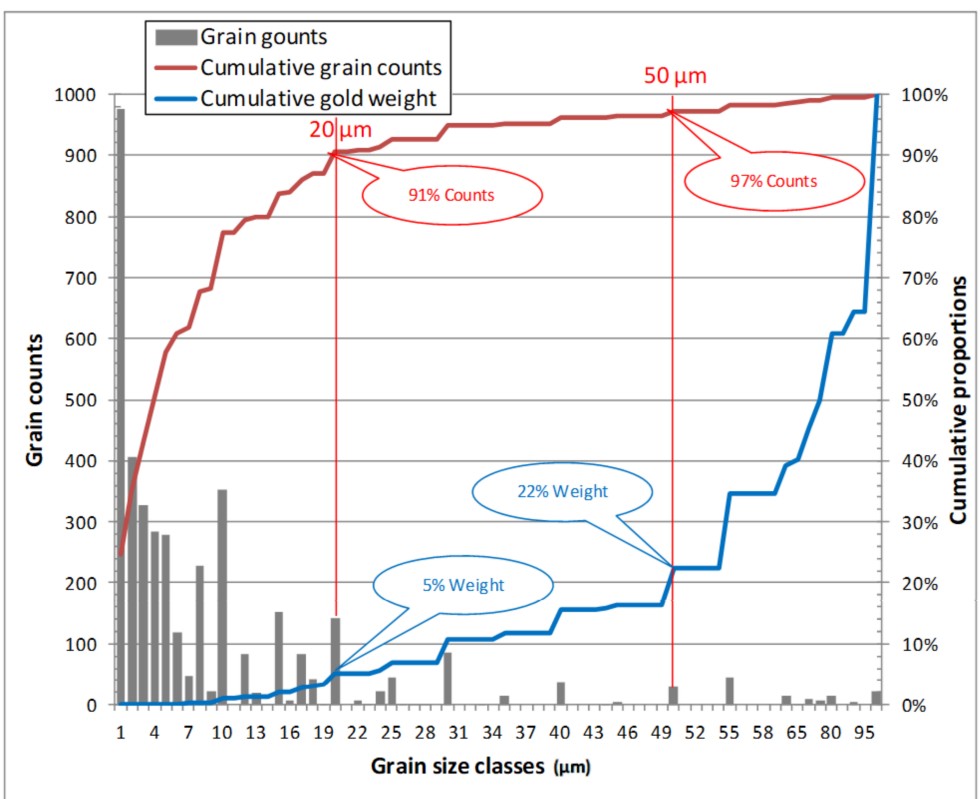

**Figure 1.** In-situ gold grain size distribution in rocks from the Archean syn-orogenic gold deposits of the Superior Craton, Northwestern Québec, and Northeastern Ontario, Canada. Measurements on 3934 gold grains smaller than 100 μm were conducted over a 20-year period by the same mineralogist (Mme Lucie Tremblay) from hundreds of polished thin sections using a reflected light petrographic microscope. Grain counts are indicated by grey bars on the right scale, while the cumulative distribution is indicated by the red curve on the right scale. The blue curve represents the cumulative proportion of gold weight calculated from the grain size, assuming a spherical shape. While the grain counts are overwhelmingly dominated by small grains, the bulk of the gold weight is constituted by large grains, which explains the good metallurgical recovery, despite the poor grain recovery of most gravimetric concentration methods. The references at 20 and 50 μm are indicated. It should be noted that the size scale is skewed for large grains, and grains larger than 100 μm were excluded from the cumulative proportions.

The grain size distribution in rocks usually follows a log-normal distribution [60–63]. The gold grain size distribution is expected to follow such a law, which is a hypothesis that has been verified in a few deposits by the author and other groups [54]. The current grain population approximates such a distribution, with a log-average of 4.9 μm and a variation coefficient of 0.8. About 80% of the measured grains are smaller than 20 μm (longest apparent axis), which is the commonly accepted smallest size recovered in glacial sediments by conventional gravimetric methods. Furthermore, only 12% of the grains are larger than 50 μm, which represents the recovery collapse of the usual gravimetric separation methods. This grain size distribution is heavily skewed toward small grain, but this is not reflective of the gold abundance itself. The weight of contained gold in a grain is a nearly a cubic function of its size, and consequently, small grains do not account for a proportional weight of gold, despite their abundance. Since large grains are easy to recover, this leads to a fair (>80%) metallurgical recovery by gravimetric methods, despite a poor grain count recovery. Moreover, as far as mineral exploration is concerned, the information extracted from the grain is more significant than the mere weight of the gold, which is more easily obtained simply from assays.

After erosion, gold grains are dispersed in a secondary environment, along with all other minerals. In alluvial systems, small gold grains are likely to be elutriated, and the size distribution is no longer representative of the initial gold in mineralized samples. Glacial till, which essentially consists of sub-glacial grinded sediments that are not affected by hydraulic transport, is not significantly affected by the elutriation process, regardless of the transport distance [18]. Accordingly, till has been sampled for decades in gold exploration programs and regional surveys, and numerous studies [2,7,64–68] have highlighted that, like in ore deposits, most gold in till is fine-grained, with a maximum rate of occurrence of around 30–50 μm using the conventional method. However, most data collected for exploration purposes do not include an accurate size measurement, as they are limited to size bracket estimation.

### 2.1. Improvement of Concentration Methods

The efficiency of gold grain counting is dependent on the ability to recover gold from glacial sediments and correctly identify them. While the sampling of glacial sediments is a well-established protocol [69,70], sample processing still relies on artisanal techniques (e.g., hand panning and visual identification), the results of which vary between laboratories or operators and have underrated limitations and reliability issues. Many studies suggest that the conventional method used to process till samples fails to recover/identify most small grains [3,4,28–32].

In most laboratories, glacial sediments are wet sieved at 1 or 2 mm, and passing material is conveyed through a gravity separator, which is usually a shaking table with one of many designs, although some other devices, such as a centrifugal concentrator (*Falcon* or *Knelson* concentrator), spiral concentrator (*Goldhound*), elutriation columns or *Hydrosperator* [71,72], dense media separator, or various type of mechanized panner (*Superpanner*, *rotating cones*), can be used. The purpose of this operation is to wash away the light and clay-sized particles and concentrate the heavy silt- and sand-sized ones. Most of these devices can reduce the size of a sample from tens of kilograms of sediments to a few tens or hundreds of grams of heavy minerals, with a particle size ranging from 5 μm to 1–2 mm. These heavy mineral concentrates (HMC) are not suitable for efficient gold grain visual sorting, being too large and thus time consuming to sort. Based on our experience, a trained mineralogist can sort about a gram of −60 μm material per hour, while automated SEM is limited to less than 100 mg per hour. Such productivity would not be viable in a commercial routine. Consequently, in most laboratories, HMCs (or preconcentrates) require further concentration, which is typically conducted by hand panning, reducing the weight to a fraction of a gram. While gold panners can be surprisingly skilled, panning remains an artisanal process that is not parameterized and is very difficult to conduct in a rigorous and systematic manner. Since gold grains smaller than 50 μm are barely visible to a mineralogist hand lens, the panner relies solely on the presence of larger grains to evaluate his/her work. Consequently, the recovery of small grains, which is already poor using a shaking table, is susceptible of becoming anemic through the subsequent panning. Alternative solutions, such as a *Hydrospearator* elutriation device [71,72], are capable of more replicable results but are not in common usage.

The efficiency of gravity separation techniques for gold grains is influenced by a combination of intrinsic parameters of gold grains (smallness, flatness, porosity, roughness, attachments, coating, etc.) and extrinsic parameters related to the sediments that contain them (grain size distribution, bulk density, agglomeration, clay abundance, etc.). The prominent factor is the hydrophobic character of gold, since all these processes are conducted in water. An improper wetting of the surface of the grain prevents the grain from sinking, the surface tension being larger than the gravity pull on the grain. This leads to gold flotation and escape in overflows [3,4,28–32]. This issue apparently dominates the recovery collapse below 50 μm but can easily be circumvented by the addition of a wetting agent.

Shaking tables operate at their best when they are fed in a very constant manner, such as in a metallurgical circuit. Their efficiency is sensitive to a variety of parameters, such as tilt and stroke length, but it is more sensitive to the feed rate, grain size distribution of the material, and washing water rate. It is also possible that different shaking table designs (Deister, Wilfley, etc.) have different sensitivities to setting, which is an issue that has not been investigated.

Most of the tabling parameters can be controlled, except the grain size distribution. Natural till is heterogeneous by nature [41], being variously sandy or clay rich. Shaking tables work best on silty material. Large particles, such as sand, tend to interact with smaller ones, hindering their motion. Clays are typically washed away, but their abundance can change the viscosity and density of the slime. Furthermore, the grain size varies within a single sample, since the samples are allowed to settle in decanting tubs after wet sieving, and sorting may occur if a slurry mixer is used to feed the table. Thus, variations between or within samples preclude the use of fully parametrized settings and require constant readjustment by the operator. Such setting readjustments are the cause of the recovery instability, leading to gold grain counts that fluctuate [73,74].

A delicate balance exists between the production of a clean heavy mineral concentrate and achievement of a good recovery. Obtaining a clean concentrate, or a concentrate that is small but with abundant heavy minerals, requires extensive washing on the table. Such extensive washing is more susceptible to worsening the small grain non-recovery issue. Hence, maintaining a good recovery implies low concentration factors that generate large concentrations, typically in excess of 500 g, with less than 10% heavy minerals.

Improving the recovery of gold grains—more specifically, minute ones—has been achieved with the development of a set of procedures (Figure 2) on a specially designed sluice (hereafter referred as a "fluidized bed" (Figure 3)). The device is a micro-corrugated channel, in which a strictly lamellar flow of water is maintained (Figure 3a). The fluidized particle load is maintained by vibrating the sluice. The vibrations are tuned to maintain particles in suspension and trap free-running denser gold grains in the micro-corrugations, without clogging the riffles. Maintaining the particles in suspension prevents the micro-corrugations overfilling, without necessitating a vigorous water flow, and enables the efficient elutriation of light minerals in a manner that is less sensitive to grain size than in an elutriation column [71]. However, the recovery of >250 μm gold grains with the fluidized bed is not optimal due to the size of micro-corrugations. This issue is bypassed by installing the fluidized bed as a feeding apron to a shaking table, where coarse gold grains can be recovered with other large heavy minerals. Processing a 10–20 kg till sample with the device box takes approximately 40 min and produces a super-concentrate of 20–200 mg, suitable for SEM scanning without further concentration. This represents a single-pass concentration factor of 40,000×–600,000×. This operation does not require a hand panning finish, which makes it less dependent on the operators' skills. As magnetite is dense and abundant, it tends to be concentrated with gold by the optimized sluice box. The magnetic fraction of the super-concentrate is removed with a hand auto-magnet to further enhance the concentration factor.

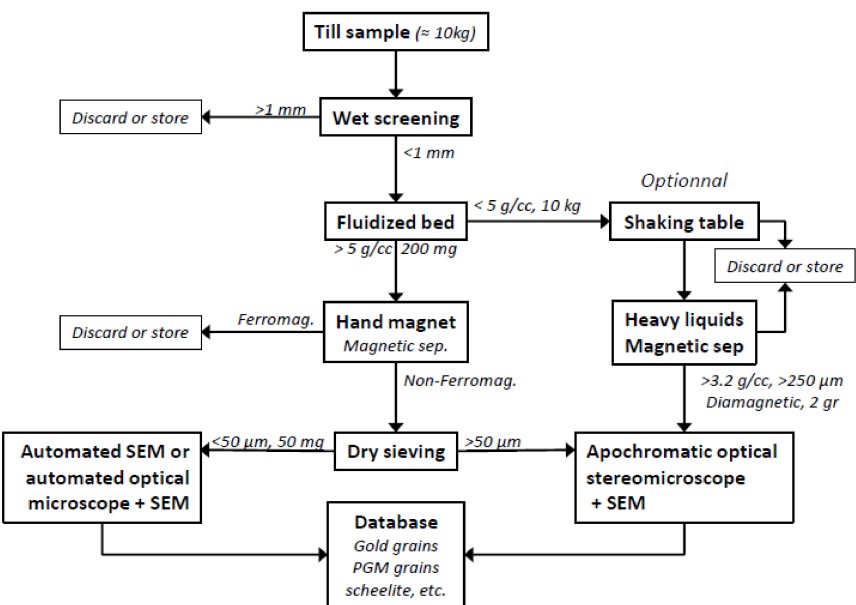

**Figure 2.** Schematized flow chart of the till sample processing of the improved method. It should be noted that the optimized sluice box can be placed as a feeding apron to a shaking table to recover gold grains or other heavy minerals larger than 250 μm.

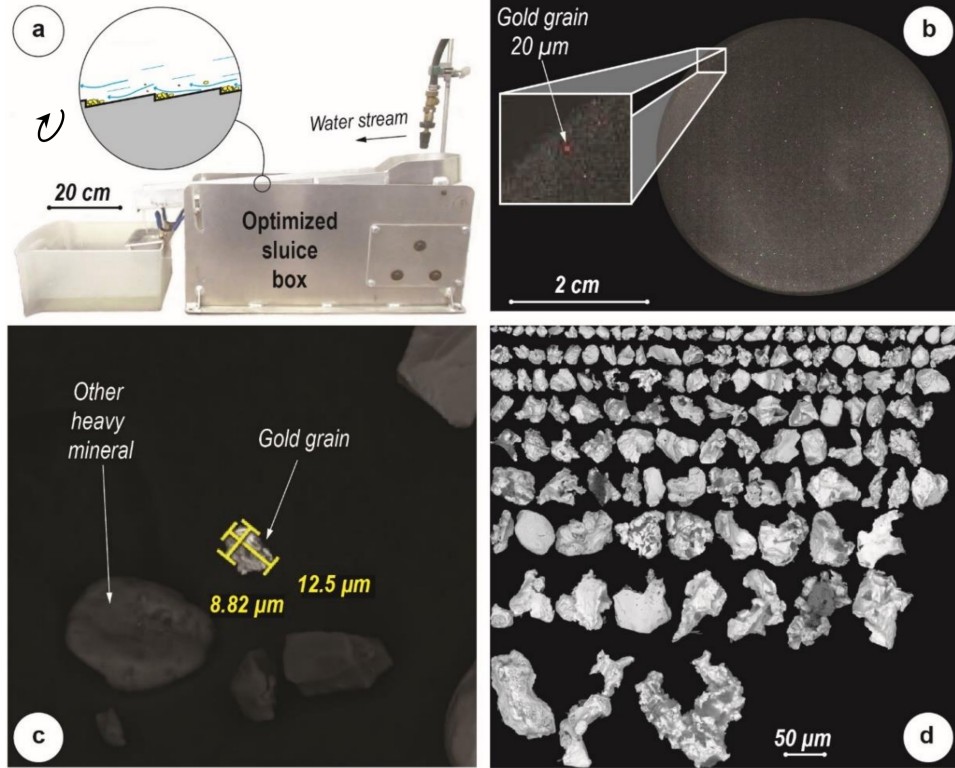

**Figure 3.** Key features of the improved method. (**a**) The fluidized bed used to concentrate the grains (the cartoon illustrates the behavior of gold grains in the lamellar hydraulic flow and elliptical motion of the bed). (**b**) Example of a custom-made holder dusted with the material that was −50 μm (bright spots are the heaviest minerals). (**c**) Example of a high-magnification back-scattered electron image of a gold grain (among other heavy minerals) used for size measurement. (**d**) Example of an automated back-scattered electron image mosaic of gold grains recovered from a single till sample. Gold grains as small as 2 μm in equivalent diameter can be recovered.

The proper preparation of the material cannot be underrated. Prior to being fed to the fluidized bed, the sample needs to be slurred with water and a wetting agent. Deflocculant is usually not required to process glacial sediments, as they do not contain significant clays (phyllosilicates). Then, the slurry is fed into a stack of sieves mounted on mechanized units, screening the >1 mm fraction. Coarse particles must be removed, since they will cause turbulences in the fluidized bed or shaking table, hindering proper separation. Sifting the material in the field is not recommended, since it would likely cause a loss of fine gold grains due to elutriation or improper wetting. Additionally, the sampled material must be free of iron or carbonate coating to ensure the proper liberation of the grains and adequate overall density of the grains and will thus not be affected by pedogenic processes.

### 2.2. Improvement of Gold Grain Counting

The second misleading premise of the conventional approach is that gold grains are easily recognized among heavy mineral concentrates. While recognizing large gold grains is easy, even for untrained geologists, properly identifying tiny specks is a matter of endless debate. A typical 10–16× triplex mineralogist hand lens enables a resolution of a few tens of micrometers, thus limiting the reliable identification of gold grain to about 50 μm, which is the size of a human hair. However, heavy minerals cannot be practically sorted using a hand lens, and the task should be carried out under stereomicroscope.

Regular stereomicroscopes, equipped with planachromatic objectives, enable a magnification of up to 40×, at which they are plagued with persistent chromatic aberration and a limited depth of field, meaning that they are usually operated at 16× to 25×. Regardless of the perceived quality of the optic, identifying gold grains that are <50 μm with such microscopes remains a tricky task (Figure 4). High-end apochromatic stereomicroscopes offer a better resolution, without limiting the depth of field, enabling a higher magnification of up to 100×. Working at such magnification limits the field of view and the depth of field, thus slowing the sorting process, not to mention the difficulty of manipulating the grains and the fatigues it causes to technicians. Tests made by trained mineralogists sorting the same concentrates using a high-end apochromatic stereomicroscope, compared to the usual planachromatic stereomicroscope, showed that the gold grain counts were increased by 57% for grains in excess of 50 um (Figure 5). While some laboratories routinely report gold grains in the 20 μm size range, a verification conducted using an SEM on such grains indicated high (up to 70%) misidentification rates, with brass chips and sulphides mistaken for gold, which is likely due to the poor resolution.

Visual identification relies on the skills of the operator and day-to-day constancies may vary over time. The development of operator-independent procedures is thus essential for yielding consistent results that can be compared over time.

Gold grain counting can be automated either on a motorized high-magnification optical microscope or with a motorized scanning electron microscope (SEM). However, any grains identified by the optical method would still require EDS analysis with an SEM to be confirmed.

Grains cannot be manipulated in the course of an automated scanning process, meaning that they cannot be swept from a pile and need to be spread as a monolayer on a stable observation substratum. In such a monolayer, large grains cannot be observed simultaneously with small ones, first, because of the limited depth of field, which would put either large or small grains out of focus, and second, because small grains may be shadowed by larger ones. Consequently, the +50 μm and −50 μm size fractions of the super concentrates are separated with disposable woven meshes in custom-made glass sieves (Figure 2). The +50 μm material is then sorted visually under an apochromatic stereomicroscope (Leica M205C (Leica Microsystems, Wetzlar, Germany)) at a magnification of up to 104× by a trained mineralogist. The −50 μm fraction is dusted on a double-sided carbon tape stuck on top of a 40 mm square aluminum plate (Figure 3b). Approximately 50 mg of the finer material (−50 μm) can be spread as a monolayer, which represents in

excess of 1 million grains. Plates are then mounted on a sample shuttle and inserted into a numerically controlled SEM.

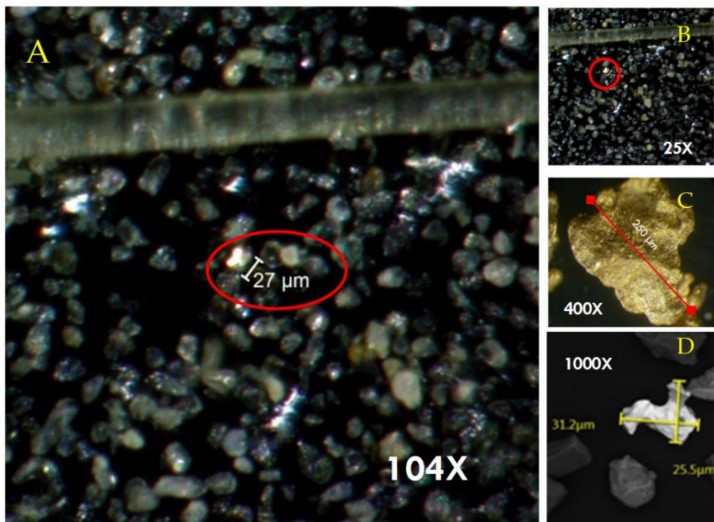

**Figure 4.** Examples of gold grains, as seen at various magnifications with different types of microscope. (**A**) 27 μm gold grain seen at 104× with a high-end apochromatic stereomicroscope (Leica M205-c, annular illumination). The blurred horizontal bar is a human hair. Tests indicate that sorting with such a stereomicroscope increases the gold grain counts by about 57%, compared to the usual stereomicroscopes. (**B**) Same field of view with a conventional plan-achromatic stereomicroscope (Leica MS-5, oblique illumination) at 25×. The 27 μm gold grain is barely recognizable, although such equipment is in current usage in most laboratories. (**C**) An example of a large 250 μm grain seen with a metallographic dark-field microscope at 400× (Wild M21). The limited depth of field does not enable the proper focus of the edges, despite the grain being a flat flake. (**D**) Back-scattered electron image of a complex 31 μm gold grain, acquired with an automated SEM routine. The depth of field and resolution of minute details, enabling textural studies, should be noted. Such images were acquired for every single grain using the automated routine. The scale bars were added manually.

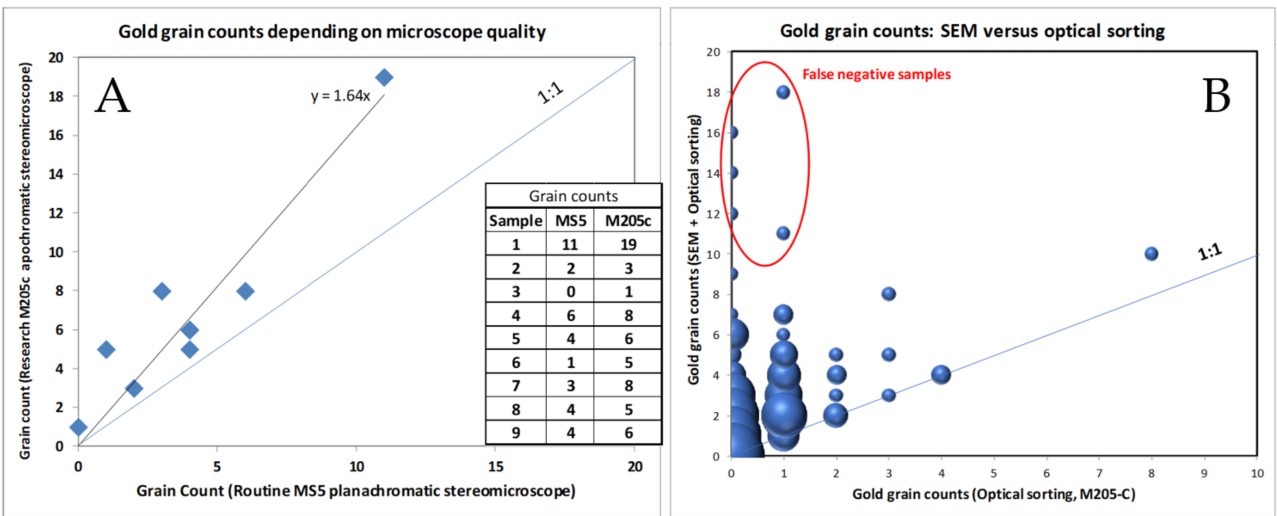

**Figure 5.** (**A**) Results of a test where the same series of concentrates were sorted for gold grain under a routine Leica MS5 stereomicroscope and a research grade Leica M205c stereomicroscope by experienced mineralogists. (**B**) Results of a test on 134 samples that were sorted visually with a research grade Leica M205C stereomicroscope and reprocessed with SEM scanning. Only 20% of the grains were noted by visual sorting, and 5 out of 6 anomalous samples were missed. The bubble size refers to the number of samples (1 to 23) with a specific count.

The purpose of the analysis is to count all the grains, regardless of the initial sample weight. Therefore, all grains have to be exposed to the electron beam. Mixing the material into epoxy slurry, which is commonly done for metallurgical testing, means that only a fraction of the grains would be exposed by cutting and polishing. To overcome this, the material is dusted onto a conductive substratum, rather than poured into epoxy. Spreading the material into a monolayer to be cast in epoxy would create serious issues in polishing, considering the smallness of the grains (10–50 μm) versus the large area of the mounts (4 cm). The slightest polishing error would either leave unexposed grains or erode them completely. Since no sample material is usually left, such an error would ruin the sample. Furthermore, with a grain size between 10 and 50 μm, it would be impossible to intersect both small and large grains by polishing. Trying to intersect large grains on their center (25 μm above the substratum) would leave small grains unexposed and trying to intersect small grain would cut the large one close to their base, thus causing size underestimation.

The use of dusted material also has the advantages of exposing the entire outline of the grain, enabling an accurate size measurement and the preservation of the surface texture of the grains, which is used for shape classification [41]. However, the procedure has drawbacks, mostly relating to EDS analysis [35,42].

Scanning for gold grain can be accomplished on most numerically controlled and motorized SEM. The procedure presented here was implemented on both a 2013 Zeiss EVO MA-15-HD (Carl Zeiss AG., Cambridge, England) with a $LaB_6$ emitting source, equipped with an Oxford Instrument X-Max 150 $mm^2$ EDS-SDD detector (Oxford Instruments plc, Abingdon, England), and a 2018 Zeiss Sigma 300 VP FEG (Carl Zeiss AG., Cambridge, England), equipped with Oxford Instrument Ultim-Max 170 $mm^2$ EDS-SDD detector (Oxford Instruments plc, Abingdon, England). The analyses are performed under a low nitrogen pressure of 40 Pa in the sample chamber to limit outgazing of the substratum but allowing for the drainage of the electrons. Up to 8 mounts plus calibration material can be inserted on a custom-built stage at the same time. The routine, which has some similarities with the RPS protocol (rare phase search) of an MLA (Mineral Liberation Analyzer [75]), is implemented on Oxford Instruments' Aztec-Feature platform (version 4.2). The routine acquires a backscattered electron (BSE) mosaic of 520 to 560 images of 1.92 × 1.44 mm (1024 × 768 pixels, resolution of 1.88 μm/pixel), encompassing the entire surface of the sample mounts (Figure 4a). The BSE brightness and contrast are adjusted, so particles with a density >5 $g/cm^3$ are discriminated. This density is used to avoid detecting monazite, which is ubiquitous (thousands) in the HMC and hinders the length of the acquisition. Using a density lower than that of pure gold (i.e., 17 $g/cm^3$) allows for the detection of gold grains, even if they are shadowed or have irregular surfaces. Clusters of bright pixels are then segmented into individual particles, and the EDS acquires an X-ray spectrum that is deconvoluted into a qualitative chemical analysis. The mineral specie is identified in real time with a classification tree, highlighted in false color on the mosaic. Upon the completion of the automation, a high resolution (image dimensions of 144 × 108 μm, 0.14 μm/pixel) is acquired for each gold grain, and a semi-quantitative EDS spot analysis is prompted. The size and detailed morphological characteristics (long and short axis, axis ratio, equivalent circle diameter, and area) are measured. All the information is then compiled and presented in a report. Figure 3d shows a mosaic of gold grain images from a single till sample, which is automatically generated by the system. The automated routine has the benefit of not being labor intensive, running overnight without attendance, and not relying on the operator's skills or day-to-day conditions. Random verifications indicated a reproducibility rate in excess of 99%.

Compared to the RPS protocol of the MLA or TMS protocol of the QEMScan, the current routine achieves a lower speed of acquisition, assuming a similar resolution and settings. However, it provides fully deconvoluted EDS analyses, instead of proxies based on matching the spectrum with a library with information relevant to mineral exploration [42]. High-resolution BSE images required for shape analysis [41] can be assembled into a mosaic, just as grains in false color can be assembled for the MLA and QEMScan. The routine can be

programmed on any particle analyzing software that is flexible enough for programming custom routines, such as Zeiss' Mineralogic, but not on an MLA or QEMScan, since these do not deconvolute the EDS spectrums into real analysis. However, since the routine does not require phase relationship analysis, it was implemented on general-purpose particle analysis software (Oxford Instrument Aztec-Feature) and could have been implemented on most other similar systems (Zeiss SmartPI, etc.).

Gold grain counts obtained by an SEM-based routine have been compared to those of visual sorting. Fluidized bed super-concentrates from 133 samples were visually sorted by an experienced (+25 years) mineralogist using an apochromatic stereomicroscope (Leica M205-C) over a two-year period. Approximately 15% of the super-concentrates were re-sorted by a second mineralogist as quality control, including every sample with more than 3 grains. Sorted gold grains were extracted and checked with the SEM, including high-magnification BSE images and EDS analysis. Following the automation of the SEM-routine, the 133 super-concentrates were re-processed. A total of 77 grains were initially extracted by visual sorting, while 303 more grains were detected by SEM, of which 20 were larger than 50 µm. This means that only 20.3% of the grains were seen in the course of initial visual sorting, despite the quality of the optics and the experience of the mineralogist (Figure 5b). The visual sorting failed to detect the abundance of grains in 5 of the 6 anomalous samples (>10 random grains for this specific project). Failing to detect an anomalous samples would of a concern in regard of mineral exploration.

An automated optical pre-sorting of gold grain can be inserted in the procedure in order to reduce the SEM time. The distinctive yellow color of gold is due to its quite distinctive reflectance spectrum, which is highly reflective from red to green but poor for the blue wavelength [76,77]. Such a spectral signature can be recognized with a standard RGB camera, and pixels with such a signature can be readily detected by subtracting the signal of the blue pixels from that of the red pixels. However, this technique, based on only three wavelengths, is not sensitive enough to efficiently discriminate gold from other yellow minerals, such as sulphides, monazite, some garnets, brass, etc. Furthermore, at a high magnification, fringes of yellow chromatic aberrations may rim the quartz and feldspar grains, which confuse the system. Therefore, gold grains cannot be detected on the premise solely of their distinctive color. A single sample could generate tens of thousands of such yellow grains or false positives. The issue can be circumvented by reprocessing the segmented images of yellowish grains with a trained dataset of confirmed gold using a deep convoluted neural network (Inception V4 architecture) [78–80]. By properly training the system, the false positive rate can be reduced to a few tens per samples. In practice, the super-concentrates are mounted as in SEM scanning and placed onto a sample shuttle clipped to a motorized microscope stage (Zeiss AxioZoom V16, equipped with an Episcopal LED ring light, a PlanNeofluar 2.3× objective, and a Zeiss AxioCam 506 color digital camera (Carl Zeiss AG., Oberkochem, Germany) operated at 63×. A mosaic of the image covering the sample, with a resolution of 1.2 µm and a depth of field of 28 µm, is assembled in image processing software (Zeiss Zen 2.5 Pro). Images of the grains are extracted, along with their coordinates, and processed using the CNN (ARTPhot) routine. Then, the sample shuttle is transferred to the SEM, along with the possible gold grain coordinates. Grains are then brought one by one under the SEM hyperconical lens, where the high-magnification BSE image and EDS spectrum are acquired to confirm the grains.

Optical presorting has the benefits of being fast and cost-effective. Tests conducted in autumn 2018 indicated that more than 95% of the gold grains detected by SEM scanning were also detected by automated optical sorting (i.e., a 5% false negative rate), a performance that improves through time with the retraining of the AI routine. Again, while the detection rate is near complete with a gold grain larger than 25 µm, it falls to about 50% in the 5–10 µm size range. Contrary to BSE-EDS scanning, the currently available optical sorting technique does not enable the detection of other meaningful minerals. The capability of the automated optical sorting to detect gold grains at a similar rate to SEM scanning

suggests that the failure to detect them by visual sorting is not caused by an insufficient optical resolution, but rather the limited ability of the mineralogists to recognize them.

### 2.3. Recovery Measurements

Accurately measuring the weight recovery of gold by the gravimetric method is achieved simply by assaying and mass balancing the feed, the concentrate, and the tails. Conversely, measuring recovery in terms of grain abundance is difficult, since the initial abundance of grains in the feed cannot be measured without counting them. Gold grains are too small to be efficiently manipulated, so it is complicated to manufacture a synthetic reference sample with a pre-established number of grains [73], unless only large grains are used, which are notoriously easy to recover, and this introduces a bias. The issue can be circumvented by reprocessing the tails. Through successive reprocessing, the tails of a sample will be the feed of its successor, and the number of recovered grains will decrease according to the recovery rate, which is then a convergent factorial series. Recovery is then calculated as:

$$R = 1 - N_2/N_1,$$

where Nx is the number of grains recovered on a specific concentration cycle.

The recovery can be calculated separately for various grain size intervals, shapes, compositions, etc. Tests are routinely conducted by reprocessing the sample's tails as part of the QAQC program. A recovery curve based on grain size is calculated, suggesting a recovery greater than 90% for grains larger than 80 µm, with a progressive decrease with the grain size and a recovery collapse of around 20–25 µm (Figure 6). The recovery rate is primarily influenced by the concentration process since the identification rate with the SEM is greater than 95% even for grains of a few micrometers.

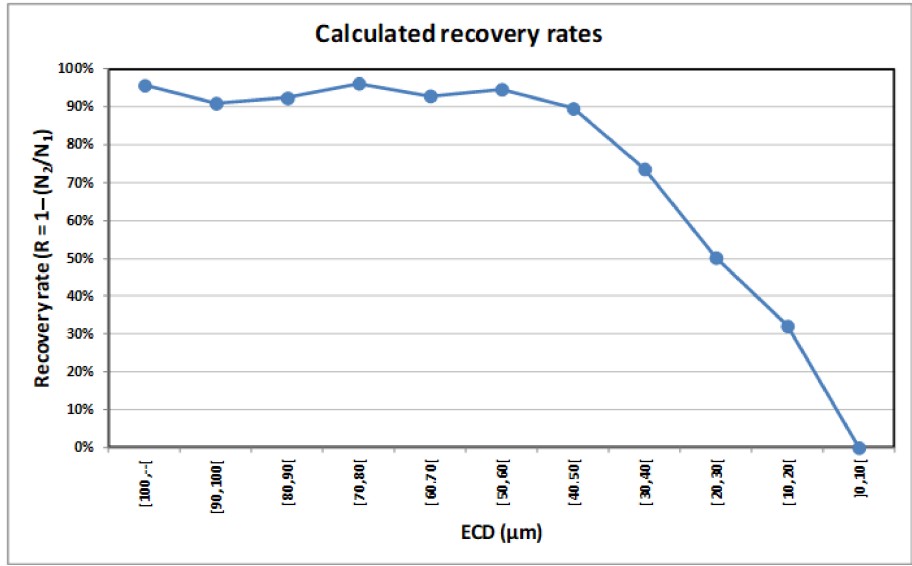

**Figure 6.** Recovery rates of gold grain with the fluidized bed according to the grain size, as obtained from the reprocessing of samples during a recent QAQC protocol. The recovery rate starts declining to below 40 µm, collapses for grains smaller than 20 µm, and fades almost to zero below 10 µm. The calculations are based on 2528 grains from 43 samples processed over an 8-month period.

### 3. Results

To illustrate the effectiveness of the improved method, we compare recent surveys conducted over the same area using both the conventional and improved methods (Table 1). The results include the counts and sizes of gold grains recovered from the till samples collected in a ca. 250 km$^2$ area surrounding the Borden gold deposit (Ontario, Canada). A first series of regional samples were collected by Ontario Geological Survey and Probes

Mines between 2011 and 2014 and processed by the same laboratory using the conventional method (Table S1a,b from Supplementary Materials). To calibrate the survey, a series of samples were also collected by Probe down-ice of the Borden gold deposit in 2014. The second series of samples, collected between 2015 and 2017 on behalf of Goldcorp Canada, includes regional surveys that partly encompass Probe surveys, plus a replicate of Probe samples down-ice of the Borden gold deposit (Table S2a–c from Supplementary Materials). Follow-up samples down-ice of anomalies and samples collected by subsequent sonic overburden drilling have been excluded. The sampling procedures and sample size and quality are assumed to be similar, with the exception that the 2014 samples were sifted in the field, while the others were sifted in laboratory conditions. The locations, survey types, and gold grain counts from the conventional and improved methods are listed in Table 1. The sizes (width and length) of the gold grains recovered by the conventional and improved methods are discussed in the companion paper [41], while their chemistry is discussed in a second companion paper [42]. The results will be discussed solely in terms of the method efficiency, and no details pertaining to mineral exploration are disclosed here.

**Table 1.** Statistics on the gold grain counts from the various surveys used in the current study.

| Survey | Type | Year | Method | Samples | Total (Grains) | Average (GG/s) | Nor. Avg. (GG/10 kg) | Maximum (Grains) |
|---|---|---|---|---|---|---|---|---|
| OGS | Regional | 2011 | Conv. | 69 | 78 | 1.13 | 1.41 | 10 |
| Probe Minerals | Regional | 2012, 2014 | Conv. | 724 | 582 | 0.80 | 1.00 | 10 |
| Probe | Borden | 2014 | Conv. | 40 | 487 | 12.17 | 15.21 | 41 |
| Goldcorp | Regional | 2015, 2016, 2017 | ARTGold | 985 | 9119 | 9.26 | 10.64 | 310 |
| Goldcorp | Borden | 2016 | ARTGold | 61 | 6647 | 105.69 | 116.2 | 507 |

### 3.1. Study Area

The Borden gold deposit is located 160 km south-west of Timmins, in the Wawa Sub-province of the Archean Superior Province. The gold deposit occurs in the west trending Borden Lake greenstone belt in the southern portion of the Kapuskasing Structural Zone [81–83]. The Borden Lake greenstone belt, which has undergone high-grade metamorphism (up to granulite facies), consists of mafic to ultramafic gneisses, pillow basalts, felsic meta-volcanic rocks, felsic porphyries, and tonalites that are overlain by a >30 m thick suite of Timiskaming-aged clastic metasedimentary rocks [84–87]. The gold mineralization, initially identified in outcrops by Probe Mines, essentially occurs in a ductile shear zone within the volcano-sedimentary horizon. It consists of a high-grade core with gold associated predominantly with quartz flooding/veining and potassic alteration and a low-grade envelope with gold associated with disseminated and fracture-controlled pyrite/pyrrhotite with local silicification [88,89]. In 2015, the deposit was acquired by Goldcorp Canada Inc. (currently Newmont-Goldcorp Corporation), and commercial production was inaugurated in September 2019 [90].

The study area has been eroded by the Wisconsinan glacial advance, which left the bedrock partly covered by Wisconsinan glacial sediment (Figures 7 and 8). The detailed surficial geology is reported in Gao [91] and Girard and Villeneuve [92]. Kilometer-sized patches of thick and continuous till are restricted to the northern and south-eastern portions of the study area, whereas thin discontinuous veneers of till occur mainly in the eastern and western portions. The till consists of poorly sorted silty sand or sandy diamicton with pebbles. The study area also contains glaciofluvial and glaciolacustrine material, deposited during the withdrawal of the late Wisconsinan icesheet, a sampling of which has been avoided. The regional ice-flow direction is oriented south–south-west, as indicated by the glacial landforms (e.g., drumlins, flutings, crag, and tail) and striations engraved on the

bedrock. This ice-flow dispersion does not seem to have reworked the former dispersion or to have been remobilized by subsequent sedimentary events.

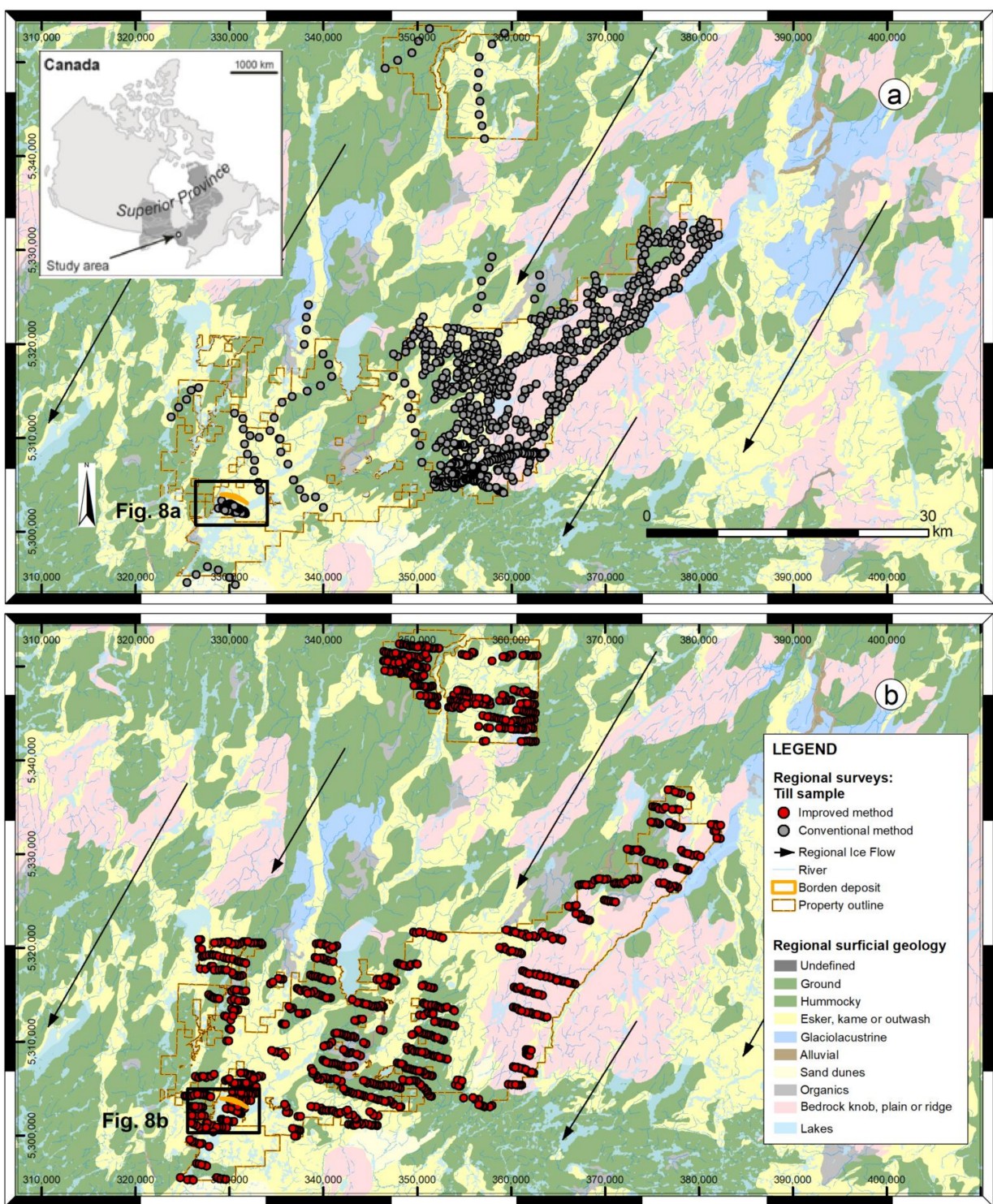

**Figure 7.** Map of the regional area surrounding the Borden gold deposit, showing a surficial geology and the locations of the till samples processed with the conventional method (**a**) and the improved method (**b**). The number of samples for the conventional method *n* = 833 and for the improved method *n* = 1192 should be noted.

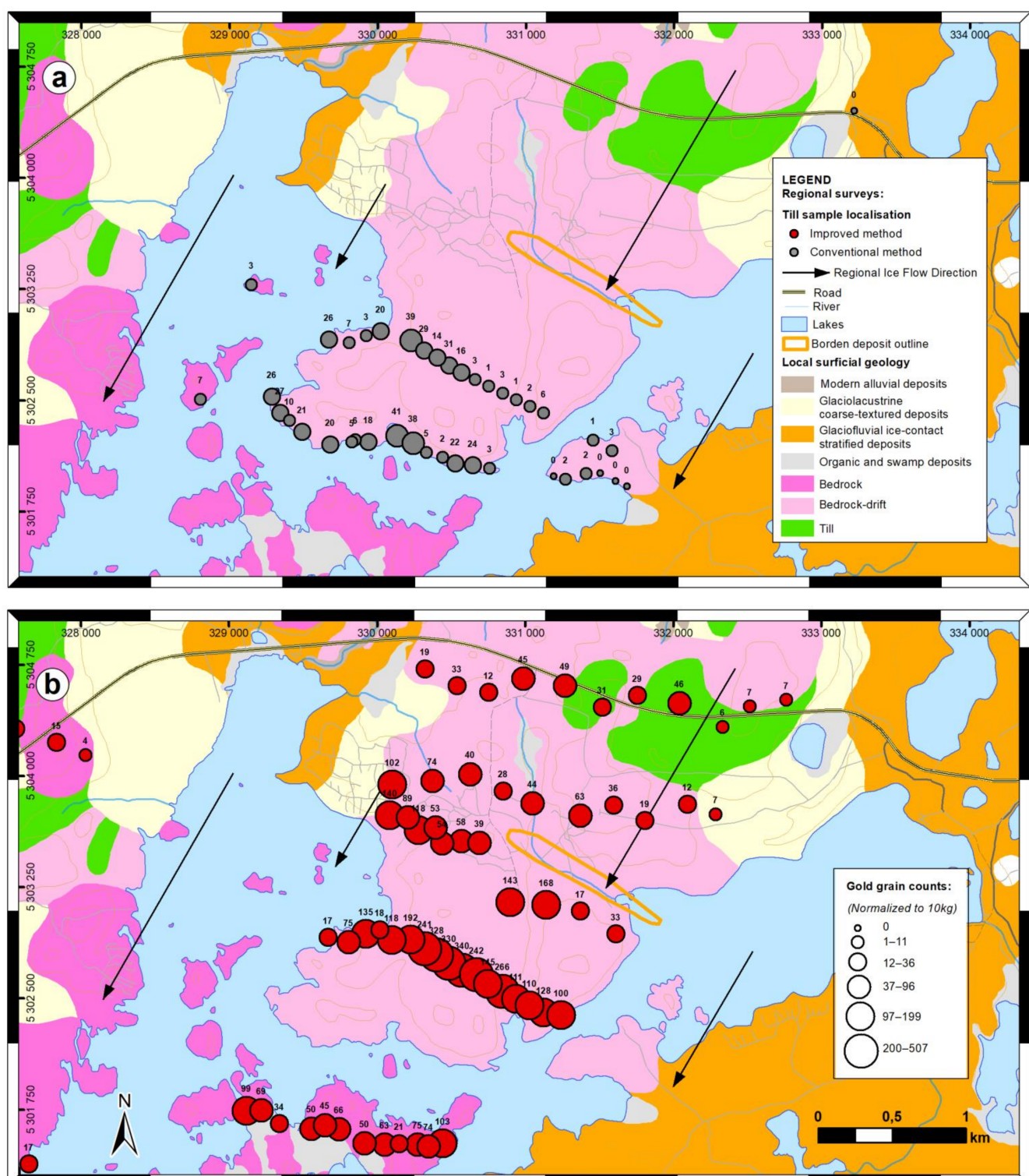

**Figure 8.** Map of the local area surrounding the Borden gold deposit, showing a surficial geology and the locations of the till samples (orientation surveys) processed with the conventional method (**a**) and the improved method (**b**). The number of samples for the conventional *n* = 40 and for the improved method *n* = 61 should be noted. The bubble diameter refers to the gold grain count normalized to a 10 kg sample. On the main profile down-ice of the deposit, samples for the improved method were collected in the same location (within GPS accuracy) from the previous conventional survey and can be considered as paired samples.

## 3.2. Till Surveys

Till samples were collected with hand shovels using the same protocol during the regional and orientation surveys. The orientation till survey, commissioned by Probe Mines prior to their regional survey in 2014, aimed at characterizing the footprint of their discovery before initiating the regional surveys. A total of 40 samples were collected along 2 profiles located 1 and 1.5 km down-ice of the outcropping mineralization (Figure 8). The survey did not extend southward due to the presence of the Borden Lake coinciding with the property boundaries. The samples were processed using the conventional method, and the results indicate a dispersion train about twice the width of the outcropping occurrence, with a maximum count of 41 gold grains in a sample collected 1.5 km down-ice. This is contrasting with the results for the 793 samples from regional surveys, either by the Ontario Geological Survey (OGS, 2011, 69 samples; [93]) or Probe in 2012 and 2014, which yielded a total of 660 grains [no report available], with an average count of 0.83 gold grains per sample using the conventional method.

After its acquisition of the Borden property from Probe Mines, Goldcorp commissioned a series of surveys between 2015 and 2017 [94–97]. A regional till survey using a sample spacing similar to Probe Mines' regional survey was conducted to cover previously inadequately sampled areas. A total of 985 regional samples were collected and processed using the described method, yielding an average count of 9.3 grains per sample. In addition, an orientation survey was conducted to assess the extent of the mineralization footprint, with 61 samples collected along 6 profiles located from 2 km up-ice to 3 km down-ice of the mineralization. The profile located 1 km down ice of the mineralization replicated the one from Probe Mines, collecting the samples at the exact same sites (within GPS accuracy) and therefore generating 15 duplicates. The maximum counts, normalized to 10 kg samples, down-ice of the mineralization are of 185 gold grains at 200 m, 479 gold grains at 1 km (compared to the maximum count of 39 of the Probe survey along the same profile), 118 grains at 2 km, 70 grains at 4 km, and 30 grains at 6 km, with counts below 30 grains further away. Such a pattern is typical of glacial dispersion trains rooted directly on a mineralized occurrence.

## 3.3. Gold Grain Counts

For the collected till samples, gold grain counts using the standard and new methods differed markedly. Overall (regional plus orientation surveys), the conventional method recovered 1147 gold grains from 833 till samples (~6436 kg of −2 mm material, assuming 8 kg per sample), with an average of 0.178 grains per kilogram. By comparison, the improved method recovered 15,566 gold grains from 1046 till samples (~11,873 kg, or 8252 kg −1 mm), with an average of 1.68 grains per kilogram. Thus, the improved method recovered 9.43× more gold grains per kilogram of sifted material, compared to the conventional method. This leads to significant differences in the count distribution (Figure 9). The conventional method failed to recover any gold grain from 423 samples (50.8%), whereas the improved method recovered gold grains in all but 18 samples (i.e., 1.65% of samples were deemed barren). Similarly, for the regional survey, the conventional method yielded only four samples with an excess of 6 grains, with a maximum count of 10, meaning that nearly all samples had counts lower than the average of the new method. Similar ratios (7–10×) between the counts from the two methods were obtained in various proficiency tests during the other surveys.

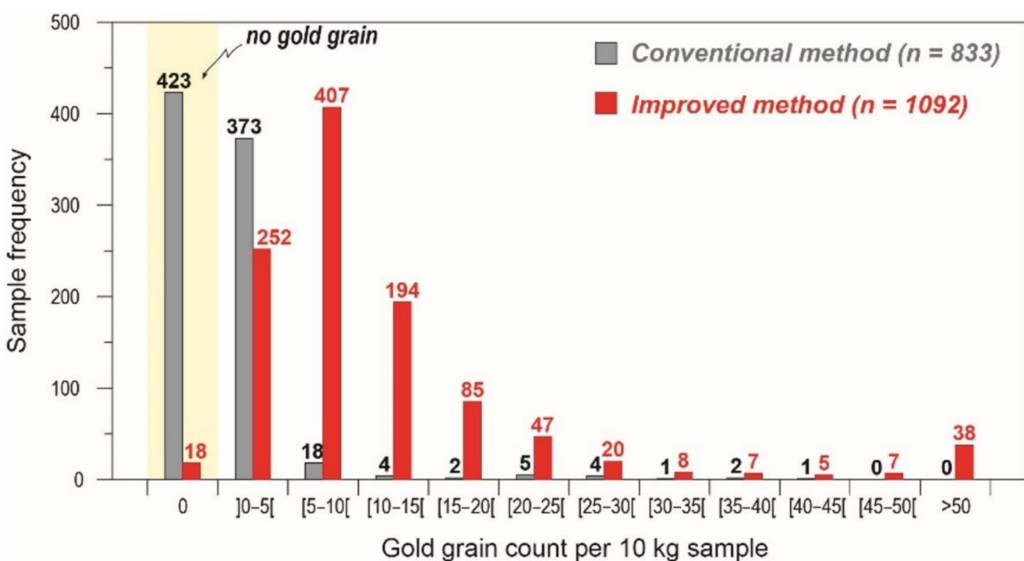

**Figure 9.** Comparison of the gold grain count per sample (normalized to 10 kg) obtained for the conventional and improved methods from till surrounding the Borden gold deposit.

As previously mentioned, SEM scanning detected approximately 5× more grains than visual sorting using a research-grade stereomicroscope, which itself detected 1.7× more grains than sorting under a routine stereomicroscope. While not rigorously tested, SEM scanning seems to yield approximately 8.5× more grains than visual sorting using the conventional method. This suggests that a large part of the improvement of the new method is not related to the increased grain recovery, but rather to the improved counting. Recovery of a fluidized bed may be similar to the recovery of a shaking table, since the estimation of the latter may be impaired by the deficient counting technique. From an operational standpoint, the fluidized bed still has the benefits of directly producing a super-concentrate without hand panning.

*3.4. Gold Grain Sizes*

Since the conventional method is considered to be efficient in recovering grains larger than 100 μm, the increase in grain counts will be primarily from detecting more small grains, assuming a constant sample weight. While the automated SEM method generates an accurate measurement of grain dimensions, the conventional method only provides visual estimates typically disclosed as 25 μm increments. To compare the results, grains abundances as measured by SEM were pooled using the size interval typically used for the conventional method, with samples normalized to 10 kg and sieved at 1 mm (Figure 10). Given the large differences in abundance between samples from the orientation and regional surveys, the results are presented separately. For the conventional method, the regional samples (*n* = 793) represented 6145.54 kg of material, while the orientation samples (*n* = 40) comprised 290.6 kg, with 660 and 487 gold grains, respectively. For the SEM-based method, the regional samples (*n* = 985) represented 11,122 kg of material, and the remaining orientation samples (*n* = 61) represented 751 kg of material, with 9119 and 6447 gold grains, respectively.

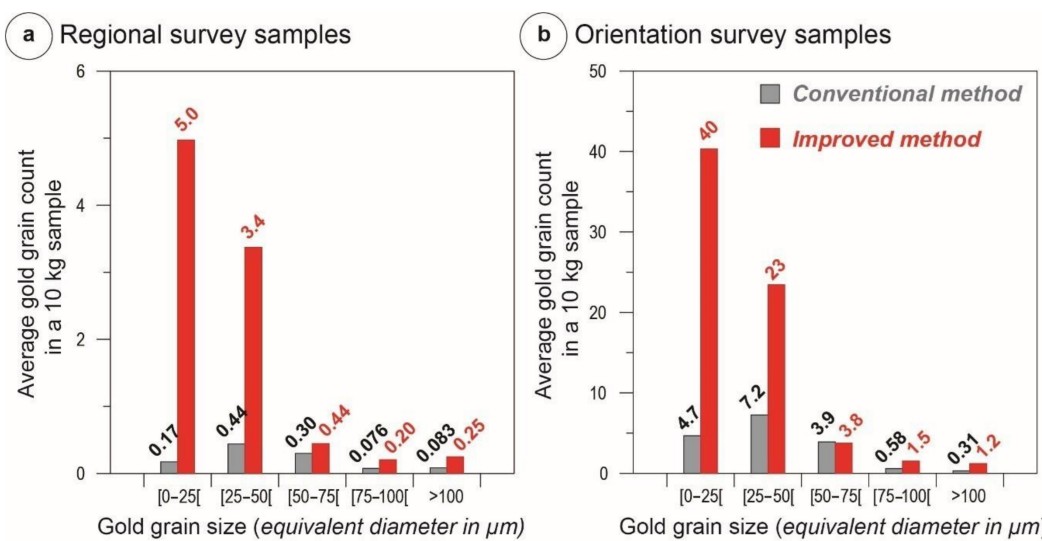

**Figure 10.** Comparison of the size distribution of the average gold grain count in a 10 kg (−1 mm) sample for the conventional and improved methods. Gold grain size distribution of (**a**) regional till samples and (**b**) orientation till samples collected around the Borden gold deposit. See the text for further details. It should be noted that the vertical scale for the orientation samples (**b**) is 8.3× the vertical scale of the regional samples (**a**).

In the regional samples, the improved method recovered 28.6× and 7.67× more gold grains than the conventional method for the [0–25[ μm and [25–50[ μm size ranges, respectively (Figure 9). For the [50–75[ μm, [75–100[ μm, and >100 μm size ranges, the improved method recovered 1.48×, 2.67×, and 3.02× more gold grains relative to the conventional method, respectively (Figure 9). In the orientation samples, the improved method recovered 8.61× and 3.23× more gold grains than the conventional method for the [0–25[ μm and [25–50[ μm size ranges, respectively (Figure 9). However, for the [50–75[ μm size range, the conventional method recovered 1.04× more gold grains than the improved method (Figure 10b), indicating approximately the same recovery rate for the two methods. For the [75–100[ μm and >100 μm size ranges, the improved method recovered 2.63× and 3.90× more gold grains than the conventional method, but such results can be misleading considering the small number of counts (Figure 10b). Thus, the improved method generally recovered more gold grains than the conventional method across the range of grain sizes for both the regional and orientation samples. More importantly, as the gold grain size decreases, the difference in performance between the two methods increases, since the improved method is far more efficient at recovering fine gold grains (<25 μm), especially from regional samples.

*3.5. Paired Samples*

To confirm that the observed differences for gold grain counts and sizing between the two methods are not related to till heterogeneity, the results from 15 paired samples collected 1.5 km down-ice of the Borden gold deposit were compared (Figure 11). The counts in these samples are elevated (up to 48 for the conventional method and up to 340 for the improved method, with both normalized to 10 kg), thus enabling statistical comparison. Aside from the two pairs of samples, the improved method recovered 5×–100× more gold grains per sample than the conventional method (Figure 10a). A total of 201 gold grains were detected by the conventional method (i.e., 17 grains per 10 kg of material), whereas the improved method yielded a total of 3356 gold grains (226 grains per 10 kg of material). This observation indicates that along the profile of the paired samples, the improved method recovered 13.4× more gold grains per weight of material. The results also show differences for the various size fractions. The improved method recovered 16.7×, 21.8×, 5.8×, 3.8×, and 12.8× more gold grains than the conventional method for

the [0–25[ μm, [25–50[ μm, [50–75[ μm, [75–100[ μm, and >100 μm size ranges, respectively (Figure 10b). The difference in counts between the two methods is stark, and a paired Student test indicates a probability of $1.6 \times 10^{-5}$ of representing the same population.

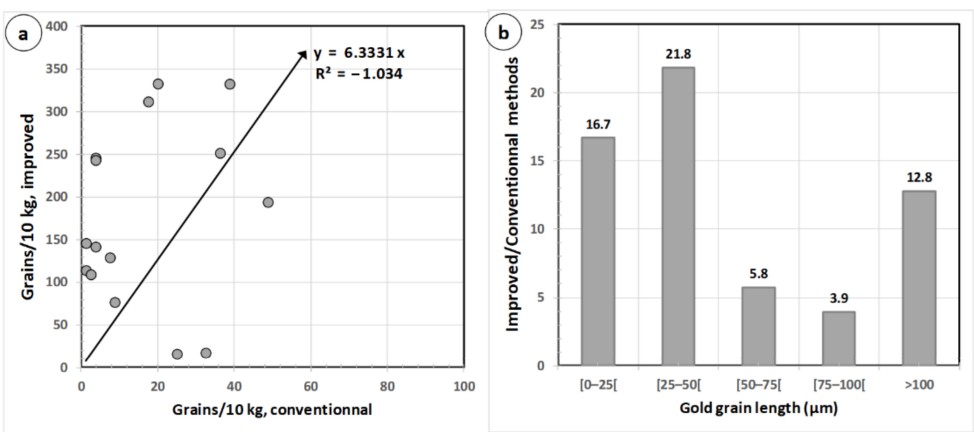

**Figure 11.** (**a**) Comparison of the conventional and improved methods in gold grain counts per sample for 15 paired samples. (**b**) Comparison of the ratio of the number of grains per 10 kg for the improved versus the conventional methods, according to grain size brackets.

*3.6. Benefit of Improving the Counts*

If gold grains are scattered in a homogeneous medium (a till in a specific sampling site), the probabilities of them occurring in an aliquot (sample) follows a Poisson distribution (or binomial distribution if using a limited number of aliquots). Hence, the probability of having a certain grain in a sample equals the probability of having any other grain in the same sample [72,76] and is equal to the probability of having this specific grain in another aliquot. This means that the number of grains present in a set of samples taken from a homogeneous site will differ from sample to sample in a predictable manner, according to the mass function. Assuming an average number of grain per sample (expectancy; $\lambda$, which, in a Poisson distribution, is equal to the variance), the distribution will have a standard deviation of $\sqrt{\lambda}$ and a variation coefficient of $1/\sqrt{\lambda}$. This means that the smaller the average number of grains in a sample, the higher the variability of the counts in the sample. Thus, for an average of 1 grain per sample, as seen in many regional surveys, the standard deviation would be 1, which is equal to the average count. Then, 36.8% of samples would be barren, and samples with 6 grains or more would occur in 1 out of 1000 samples, without necessitating the contribution of a local secondary source (mineral occurrence) and being considered anomalous.

On a regional scale, the till blanket is not necessarily homogeneous, with a mix of contributions from distant and local sources. Thus, the expectancy of each sample is different, and the distribution of such expectancies would follow a distribution that is superposed onto the intrinsic variation induced by the Poisson mass function on each individual sample. The variance of such a system is then equal to the sum of the variance of each probability function affecting the system. This increases the variance in the gold grain abundance among the samples, thus increasing the odds of having barren samples and non-anomalous elevated counts.

The regional population, as measured using the conventional method, yielded an average of 0.83 grains per sample, with a standard deviation of 1.24 grains and a variation coefficient of 1.5. Assuming a Poisson distribution, this means that 43.6% of the samples are expected to be be barren, and 99% of the samples are expected to have 3 or less grains. According to the Poisson mass function, a count of 5 or more grains has a probability of 0.14% or a single sample across the current survey, while it was obtained in 14 samples. Such a skewed distribution, compared to a true Poisson distribution, is induced by the till

heterogeneity on the regional scale or simply by the presence of anomalous samples tapping a local gold occurrence. Thus, a 5-grain count can be used as the anomaly threshold.

Increasing the average number of grains decreases the intrinsic variation coefficient according to a square-root function. The improved method has an average of 9.26 grains, with a standard deviation of ±11.98 gains and a variation coefficient of 1.29. This, compared to the results from the conventional method, suggests an 11.2× increase in average counts, which means a 2.7× decrease of the intrinsic variation coefficient. The highest expectancy (42.8%) would be achieved by the sample with 6 to 8 grains per 10 kg, while only one sample from the entire survey is expected to be barren. Conversely, a sample with 17 or more counts are twice or more overabundant and thus have more than 50% chance of being anomalous. Such a count can thus be used as the anomaly threshold.

The purpose of improving an analytical method is, from an exploration standpoint, to improve the sensitivity of the survey (i.e., signal-to-noise ratio). Obtaining higher counts would not be less meaningful if all counts increased proportionally, without reducing their variance. The results from the anomalous duplicate samples indicate a significant improvement in the method sensitivity (Figure 12) and its capability to enhance a feeble signal. The signal-to-noise ratio (S/N) is computed by dividing the gold grain counts of anomalous samples by the anomaly threshold obtained from the regional survey. The results along the test profile indicate that the mean S/N ratio of the conventional method is 2.7×, with 9 samples above the anomaly threshold. In comparison, the improved method yielded an average S/N ratio of 13.1× or four times more contrasted than the conventional method. Furthermore, none of the samples were below the anomaly threshold.

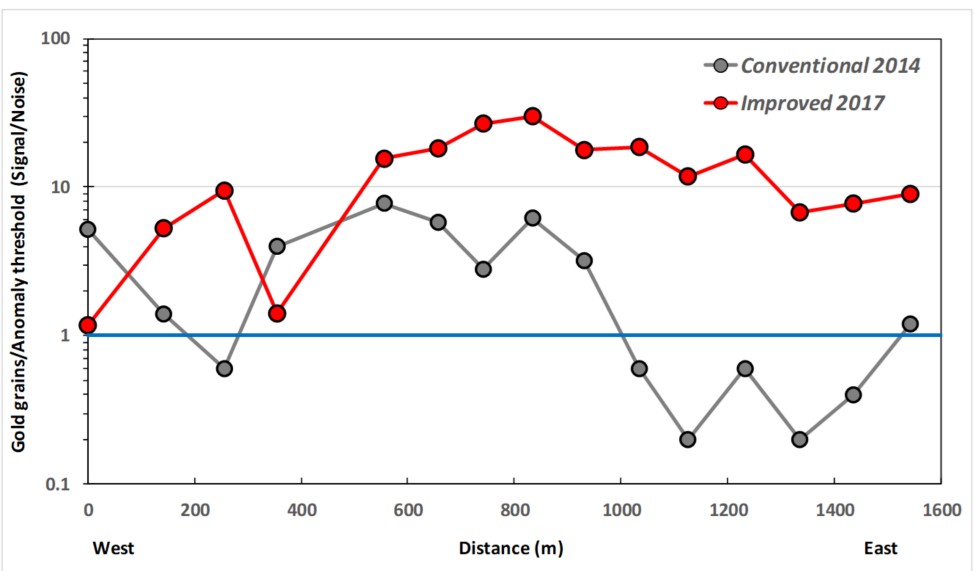

**Figure 12.** Comparison of gold grain counts normalized to the anomaly threshold for the conventional and improved methods along the profile of the 15 paired samples. It should be noted that there is a logarithmic vertical scale, while the horizontal scale represents the distance away from the westernmost sample.

Mineral deposits are defined as envelopes within which a metal is enriched above a certain grade and has the potential of being economically mined. These are typically surrounded by haloes of lower grade rocks that vanish into the barren host rock. These halos, where they are associated with the alteration system, can be significantly larger than the deposit itself. Exploration methods that are sensitive enough will be capable of detecting the signature of such halos and not only the deposit itself. Thus, the footprint of the deposit is expected to be larger than the deposit itself, and the gold grain dispersion train in the glacial sediments will also be. On the duplicated profile, the anomaly detected by the automated method encompasses 14 contiguous samples, with a minimum width of

1500 m, and remains open to the east. By comparison, only 6 contiguous samples plus one erratic sample are exceeding the background signal, with a maximum contiguous width of 700 m. The improved method can then detect the anomalous signal over at the least twice the width that the conventional method can detect it over. Thus, the detectable footprint is accordingly expanded. Similarly, the detectable dispersion is detected over a much longer distance using the improved method, although it is not accurately documented. The signal from the low-grade halos is distinctively detected, as seen from the sample collected up-ice of the deposit (Figure 7).

From an exploration standpoint, increasing the grain counts has three direct benefits: First, being capable of detecting dim signals of a broad dispersion train allows for a reduction of the sampling density. Typically, regional sampling is conducted along a fence every 1 to 5 km, depending on the ice dynamic, with a sample distance along a fence of approximately the third of the width of the expected source rock, usually in a quincunxes pattern. Widening the detectable source thus leads to a reduced sampling density along the fences. The breadth of the anomaly along the duplicated profile with the improved method thus signifies that half the samples would have been sufficient to detect the deposit during a regional survey, compared to the conventional methods. In the same way, line spacing can also be increased, thus further reducing the required number of samples. Conversely, in a case where the source is smaller than the sampling interval, increasing the sensitivity broadens the source and thus increases the odds of detecting its dispersion. In both cases, it represents either an overall cost reduction of the survey or an improvement of the odds discovery.

Second, reducing the intrinsic variance of the counts on a specific sample means that the counts are more reliable. Thus, an elevated count has better odds of being a true anomaly. In a low-count conventional survey, a mildly elevated count (e.g., 5 grains in the current survey) will not be discriminating, compared to the tail of a Poisson distribution from the regional signal. This can usually be circumvented by requiring the contiguity of two such mildly elevated counts to consider the samples as real anomalies. By improving the signal-to-noise ratio either by an intrinsic variance reduction or by removing false gold grains, the discrimination of a true anomaly related to a local source, compared to a false anomaly from the background, becomes more robust. Thus, the contiguity requirement can be eliminated in most cases, and sample spacing can be increased while maintaining the same reliability.

Third, increasing the number of grains detected by kilograms of samples means that less material is required to achieve representativeness. Till sampling is expensive, especially in logistically complex or remote areas. Reducing the sample weight from the usual 10–20 kg to 3–5 kg per sample diminishes sampling and shipping costs and significantly reduces the hardship and risk of injuries for samplers. Similarly, when sampling is conducted by drilling (reverse circulation, sonic, split-spoon, triple tubes, etc.), a wide tube caliber (PQ or 12.2 cm) is typically required to recover 10 kg of material per meter. Decreasing the sample size means that either a shorter interval can be sampled, or a smaller drilling caliber can be used. This can lead to a dramatic cost reduction, since a smaller drill rig may be sufficient. With the use of a more sophisticated method enabling the recovery of sub-micrometer gold grain currently under development, statistical representativeness was achieved with a 300 g sample, which can be collected with a BQ (6 cm) diameter split-spoon, driven with a hand-portable pneumatic hammer.

## 4. Conclusions

The methods developed for recovering and counting gold grains in glacial till are recognized as the primary tools for regional gold exploration under glaciated cover, being the sole methods enabling the detection of distant signals. The new method (referred as "ARTGold™") presents significant improvements in comparison with the conventional method. The conclusions of the present study are:

1.  Gold grains in ore deposits and till are mostly fine-grained. Despite numerous successes, the conventional method, using a combination of a shaking table, hand panning, and visual sorting, for recovering gold from till does not provide optimal results, as gold grains <50 μm are poorly recovered and identified.
2.  The proposed improved method is based on an optimized recovery procedure that concentrates fine gold more effectively and an automated SEM routine that reliably detects and counts all recovered gold grains, independently of operators' skills and surrounding conditions.
3.  From the till samples collected on a regional survey near the Borden gold deposit (Ontario, Canada), the conventional method recovered an average of 1.04 gold grains per 10 kg (−1 mm) of samples, whereas the improved method recovered 10.63 grains per 10 kg (−1 mm) of samples. The grain abundance in a sample is dictated by a Poisson distribution law, which means that the intrinsic standard deviation on the count grows as a square root of the count, meaning that the higher the counts, the lower the variability, and the better the reliability of the results. This significant difference is due to the improved method being better at recovering minute gold grains.
4.  A 15-sample profile has been duplicated about 1 km down-ice from the Borden gold deposit. The samples processed with the improved method yielded an average signal-to-background ratio of 13.1×, with all but 1 sample being distinctively anomalous. This can be compared favorably with the conventional method, which had an average signal-to-background ratio of 2.7× and which detected anomalous signal in only 8 of the 15 samples.
5.  From the perspective of a regional exploration program, the improved method allows for an overall cost reduction of the program by enabling a wider sampling pattern, more dependable results on the basis of which decisions can be made, and sample weight reduction, leading to safety improvements for samplers.

**Supplementary Materials:** The following are available online at https://www.mdpi.com/2075-163X/11/4/337/s1, Table S1: Gold grain counts and measurement form till samples processed with the conventional method, Table S2: Gold grain counts, measurement and analysis of till samples processed with the improved method.

**Author Contributions:** Conceptualization, R.G.; Data curation, J.T. and A.N.; Formal analysis, R.G.; Methodology, R.G.; Software, J.T. and A.N.; Validation, J.T.; Writing—original draft, R.G.; Writing—review and editing, H.L. All authors have read and agreed to the published version of the manuscript.

**Funding:** This study was part of a larger project funded by a Québec Fonds de Recherche Nature et Technologies (FRQ-NT) grant to L.Paul Bédard, which subsidized C. Duran's work, with contributions from IOS Servives Géoscientifiques Inc. (Projet de recherche orienté en partenariat; grant number: 2015-MI-191750).

**Data Availability Statement:** The data presented in this study are available within the article or supplementary material.

**Acknowledgments:** Goldcorp Corporation is acknowledged for granting permission to initially publish the data acquired on their property (June 2018). We are grateful to L. Paul Bédard (UQAC) and Charley Duran (UQAC) for their contribution to the writing of a preliminary version (2018) of the manuscript. Sheida Makvandi, Phillipe Pagé, Patrice Villeneuve, Natacha Fournier, Karen Gagné, Mélanie Aubin, and Karine Desbiens (IOS Services Géoscientifiques Inc.) are thanked for their valuable assistance with the SEM operation, quaternary geology, database management, quality control, drawing, and editing, respectively. D.H.C. Wilton (MUN) and G. Thompson (CNA) are also thanked for their invitation to contribute to this special issue and editorial handling. The anonymous reviewers are thanked for their constructive comments that have improved the manuscript considerably.

**Conflicts of Interest:** The authors' positions are general manager, SEM operator, research scientist, and senior scientist with IOS Services Géoscientifiques Inc. They all participated in the development of the analytical procedure assessed in this study, which is commercially offered by the corporation. Newmont-Goldcorp Corporation and FRQ-NT had no role in the design of the study, the analysis, or interpretation of data, in the writing of the manuscript, nor in the decision to publish the results.

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
