# Peer review of "Automated Gold Grain Counting. Part 1: Why Counts Matter!"

_minerals, doi:10.3390/min11040337_

Round 1

Reviewer 1 Report

Please see remarks. One chapter should be added - sample preparation . You can easily select these descriptions from the existing text. Some other remarks you will find in the text.

Author Response

Gent-Madam

First, I would like to thank you for your time and effort in reviewing this paper, I do appreciate it.

I cannot find your annoted version of the manuscript (unless merged with those of the other reviewer?), so I will answer to your general comments, and will address the annotations once you uploaded these.

Point 1: ...lack of chapter describing sample preparation.

Will add details

Point 2: ...There is no mention of the forces acting on the grain in the paper.

Will add our understanding of this issue.

Point 3: There are several measuring devices on the market...

I will add a paragraph comparing our routine with the ones available on other systems . I'm quite familiar with these, having been consulted by developper of some  manufacturer.

Point 4: ... Monolayer sample for SEM identification it is the background of good analysis.

I will add one or more paragraph discussing the pro-and-cons of dusting the material versus embedding them in resin and polishing. We do use polished pucks for usual indicator mineral counting, pucks that we manufacture ourself. This issue has been investigated in quite details, and there is a rational behind our method selection. We are aware of the stereological issue, shadowing, etc, and I will discuss.

Point 5: Comparing the traditionnal methods with the proposed ones gives similar results...

If you refer traditionnal methods as optical one, results are drastically different than with SEM. I will add a figure that shows difference in grains counts between both on the same concentrates from fluidized bed. It is highly contrasting, and the scary point is that optical sorting left numerous "false negative" samples, that would mislead explorationist.

If you refer to traditionnal "polished section" versus dusting, I will discuss this issue in details. But it is not a cost/time saving issue, it is a matter of not taking the risk of missing grains.

Point 6: Actually traditionnal methods are not very welcome...

Currently, in Canada, more than 90% of the till samples collected by the industry goes to conventionnal shaking table/optical sorting. The problem is not the lack of people to do the work, it is the lack of understanding in the mineral industry.

Sincerely

Reviewer 2 Report

The article provides a very nice showcase of the application of the Automated SEM mineralogy techniques (A-SEM). It would be necessary to provide citations on some of the earlier work on using A-SEM for gold deposit studies, specific mineral searches, and general use. A lot of work has been done and published in the past so the presented method is not novel as such. The existing A-SEM alternatives which are used routinely for gold grain searches should be noted (MLA, QEMSCAN, TIMA etc.) Their use for indicator minerals and exploration studies and in general should be cited in the introduction.

Providing data on the improvement related to the sample preparation part in relation to the A-SEM analysis part will be highly beneficial.  i.e. is the higher number of identified gold grains related more to the A-SEM ability to detect small gold grains...

Only a few more comments and suggestions are outlined in the text. After addressing these I would recommend the article to be accepted.

Author Response

Sir-Madam

I first wish to thanks you for your time reviewing this paper, it is greatly appreciated. All you comments are legitimate, and I will address them seriously.

1: (page 2) Includes litterature citations on automated mineralogy techniques in general...

I will do.

2: (page 3) Do a proper litterature search, QEM, MLA, etc.

Will do. There is effectively ample litterature on these topics, specially on the metallurgy/geometallurgy side.

3: (Page 5) Cite the use of automated mineralogy (MLA, QEM, etc).

Will add a paragraph the explain the difference between our routine  andthe ones available on other systems. I'm quite familiar with these systems, having been consulted by their developper on many occasion.

4: (page 6) Is this statement (on shaking table recoveries) based on some researches.

Yes we did extensive testing which prompted us to develop the fluidized bed. Will add a paragraph on this issue.

5: Page 8:  How do you know that (commercial laboratory do not use high end microscope)

Effectively of low pertinence, will remove. However, I did visited many commercial laboratories, and never saw an high-end stereomicroscope...).

6: Page 9: Fix the image.

Effectively, a formatting issue. I will correct, and add another one which shows relation between optical versus SEM counts.

7: Page 16: Have you been able to establish how much of the improvement is linked to the automated SEM and how to the new way to make concentrate.

This will be adressed with the figure to be added on previous point. However, this is kind of a chiken-and-egg issue, since both methods had to be developed simultaneously to work. Improving the recovrey without the capability to count would remain unnoticed, while improving counting without recovery would have been useless...

Regards